# Circulating Tumour DNA and Its Prognostic Role in Management of Muscle Invasive Bladder Cancer: A Narrative Review of the Literature

**DOI:** 10.3390/biomedicines12040921

**Published:** 2024-04-21

**Authors:** Konstantinos Kapriniotis, Lazaros Tzelves, Lazaros Lazarou, Maria Mitsogianni, Iraklis Mitsogiannis

**Affiliations:** 1Department of Urology, Whipps Cross University Hospital, London E11 1NR, UK; konstantinos.kapriniotis@nhs.net; 22nd Department of Urology, Sismanogleio Hospital, National and Kapodistrian University of Athens (NKUA), 115 27 Athens, Greece; lazarou_laz@hotmail.com (L.L.); imitsog@med.uoa.gr (I.M.); 34th Department of Medical Oncology, “Hygeia” Hospital, 151 23 Athens, Greece; mitsogiannimaria@gmail.com

**Keywords:** circulating tumour deoxyribonucleic acid (DNA), ctDNA, liquid biopsies, bladder cancer, muscle invasive bladder cancer (MIBC)

## Abstract

Current management of non-metastatic muscle invasive bladder cancer (MIBC) includes radical cystectomy and cisplatin-based neoadjuvant chemotherapy (NAC), offers a 5-year survival rate of approximately 50% and is associated with significant toxicities. A growing body of evidence supports the role of liquid biopsies including circulating tumour DNA (ctDNA) as a prognostic and predictive marker that could stratify patients according to individualised risk of progression/recurrence. Detectable ctDNA levels prior to radical cystectomy have been shown to be correlated with higher risk of recurrence and worse overall prognosis after cystectomy. In addition, ctDNA status after NAC/neoadjuvant immunotherapy is predictive of the pathological response to these treatments, with persistently detectable ctDNA being associated with residual bladder tumour at cystectomy. Finally, detectable ctDNA levels post-cystectomy have been associated with disease relapse and worse disease-free (DFS) and overall survival (OS) and might identify a population with survival benefit from adjuvant immunotherapy.

## 1. Introduction

Urothelial carcinoma of the bladder—with pure or variant histologic features—is the most common type of bladder cancer (BC) in the Western world and has a variable natural history and prognosis, which mainly depend on the clinical stage and histological characteristics of the disease [1]. Approximately 25% of BC cases are muscle invasive (MIBC, pT ≥ 2) on diagnosis, whereas a smaller proportion of MIBC cases progress from previously diagnosed high-risk non-muscle invasive disease (NMIBC). Although almost all of the low risk NMIBC cases do not progress to invasive or distant disease, MIBC is characterised by an increased risk of distant metastasis and subsequent mortality [2].

The cornerstone of the treatment for the non-metastatic MIBC is radical cystectomy with lymphadenectomy and urinary diversion in surgically fit patients. Radical surgery offers 5-year survival rates of approximately 50%. Neo-adjuvant cisplatin-based chemotherapy has been utilised to reduce the risk of metastatic disease and increase survival rates and has been included in the recommendations of the EAU and AUA guidelines [3,4]. Several studies have demonstrated a small survival benefit, which ranges between 5% and 8% at 5 years [5,6]. However, only a proportion of MIBC patients are fit enough to tolerate cisplatin-based chemotherapy, which, in combination with the modest benefit, leads to underutilisation of this treatment [7]. Adjuvant cisplatin-based chemotherapy remains debatable but is generally recommended for locally advanced or node-positive disease post cystectomy if neoadjuvant chemotherapy was not administered [2,3]. More recently, neoadjuvant and adjuvant immunotherapy have been assessed in several trials with encouraging outcomes. Although immunotherapy remains currently experimental in the neoadjuvant setting, the programmed-death ligand 1 (PD-L1) inhibitor nivolumab has been approved for adjuvant use in selected high-risk cases post-radical cystectomy, particularly when cisplatin-based chemotherapy is not indicated [8].

Since neoadjuvant treatments are associated with significant toxicities and offer no benefit in a significant proportion of patients, efforts have been pointed towards identifying predictive factors of good and poor response to tailored treatments on a patient-to-patient basis. This is particularly important considering that the delivery of neoadjuvant treatment inevitably delays radical surgery, which could, in theory, lead to disease progression for cases not responding to chemotherapy. Equally, the early start of adjuvant chemotherapy or immunotherapy in high-risk cases before clinical or radiological recurrence could potentially increase survival rates [9,10].

Therefore, there is an urgent need to identify suitable biomarkers which can provide prognostic and predictive information in different stages of the treatment to allow patient-specific escalation or de-escalation of treatment. Liquid biopsies, such as circulating tumour cells or tumour cell-free nucleic acids, could serve this purpose as they can be easily obtained and longitudinally assessed in different time points of the disease [11].

## 2. Circulating Tumour DNA: Biology and Detection Methods

Cell-free nucleic acids are continuously released to circulation through various mechanisms including apoptosis, tissue necrosis and active cell secretion [12]. Cell-free DNA (cfDNA) levels have been shown to be increased in various malignancies and can be associated with the overall disease burden and prognosis in certain cancers [13,14,15]. However, increased cfDNA levels have been reported in a variety of benign conditions such as inflammatory diseases and tissue injury [16,17]. In addition, the proportion of cfDNA deriving from cancer cells (ctDNA) is typically low and can be well below 1%, particularly in cases with low tumour burden, such as in organ-confined disease or post radical surgery. Therefore, methods with high sensitivity and specificity are required to reliably detect and quantify ctDNA [18].

ctDNA has certain biological properties to make it a very promising biomarker. Due to its short half-life, it can provide a “real-time” picture of the tumour status at the time of the sample collection [19]. Therefore, it could allow synchronous assessment of the tumour burden, assessment of the response to treatment and monitoring of minimal residual disease (MRD) post-radical treatment [13,20]. There is emerging evidence that the detection of ctDNA post-radical surgery with curative intent predates radiological relapse by several months, thus offering the potential of earlier start of adjuvant therapies with potential survival benefit. On the other hand, low or undetectable levels after surgical or systemic treatment could identify patients at low risk of progression or recurrence and therefore guide towards less aggressive treatments that avoid unnecessary toxicities [21]. In addition, as ctDNA comes from both the primary tumour and metastatic/micro-metastatic sites, it can provide information regarding mutation evolution during the course of the disease, and particularly during systemic treatment, which would otherwise require repeated invasive biopsies of the primary or metastatic sites. This can be particularly important for the detection of mechanisms of resistance to chemotherapeutic agents associated with new mutations and evolving intratumoral heterogeneity [22].

Several approaches have been utilised to detect and quantify the small fraction of ctDNA within the cfDNA in peripheral blood. These can be broadly divided into tumour-informed and tumour-agnostic methods. In the former, patient-specific mutation panels are created by profiling the primary tumour. Then, these mutations are searched in the liquid biopsy samples in order to detect and quantify the fraction of ctDNA. On the contrary, in tumour-agnostic methods, cfDNA is interrogated without previous profiling of the primary tumour, based on prior knowledge of prevalent mutations on the cancer of interest [23]. For both approaches, broader sequencing methods, such as next generation sequencing (NGS), are increasingly expanding but are still associated with significant financial costs and longer turnaround times. Moreover, NGS typically requires a quite high fraction of ctDNA that is not always available in localised cancers or in assessment for MRD [24]. Narrower detection methods, such as digital polymerase-chain reaction (PCR) and targeted sequencing, have excellent sensitivity and specificity but can only detect a limited number of mutations and potentially miss new mutations during disease evolution [25,26]. Newer approaches, such as fragment size analysis, are being evaluated to bridge this gap and improve ctDNA detection. This approach utilizes differences in the DNA fragment size between cancerous and non-cancerous cell-free nucleic acids, to allow for searching within a narrower fraction within the total circulating cfDNA [27].

Following reports of ctDNA levels successfully predicting disease recurrence and guiding adjuvant treatment for various solid tumours, attention has been brought to relevant applications in urological cancers [14,28]. There has been a growing number of studies over the last decade assessing the role of ctDNA as biomarker of organ-confined and metastatic bladder cancer. In this review, we will report the studies evaluating the prognostic and predictive potential of ctDNA in non-metastatic MIBC during different treatment settings.

## 3. Methods

We carried out a narrative review of the literature, aiming to present current and emerging evidence on the prognostic and predictive role of cfDNA and ctDNA in the neoadjuvant, peri-cystectomy and adjuvant setting in patients with MIBC. We included prospective, retrospective, single-arm or comparative studies on humans. We searched the MEDLINE via Pubmed, Cochrane library and clinical trials.gov (accessed on 28 February 2024). We utilised the terms “cell-free DNA”, “cfDNA”, “circulating-tumour DNA”, “ctDNA”, “cell-free nucleic acids”, “bladder cancer” and “urothelial cancer” combined with appropriate Boolean operators as free-text and Medical Subject Headings (MESH) terms. We did not apply any restrictions in language or date of publication.

## 4. Circulating Tumour DNA (ctDNA) in the Neoadjuvant and Pre-Cystectomy Setting

Four studies published by the research team based in the Aarhus University Hospital (Denmark) investigated the prognostic role of ctDNA status before cystectomy and its predictive role in relation to cisplatin-based NAC. In one of their first studies, Christensen et al. assessed peripheral blood samples of 27 MIBC patients treated with cystectomy but not NAC and whose tumours harboured at least one of two hotspot mutations (FGFR3 and PIK3CA). It was demonstrated that higher preoperative ctDNA levels were associated with increased risk of recurrence (*p* = 0.016), worse DFS (*p* = 0.001) and worse OS (*p* = 0.018), with 89% of patients with increased ctDNA eventually having a relapse after cystectomy [29]. In a subsequent study, Christensen et al. investigated the prognostic role of ctDNA status at different time points by using a tumour-informed approach with whole-exome sequencing of the tumour sample. They assessed ctDNA levels in 68 patients with MIBC at diagnosis (before neoadjuvant chemotherapy—NAC), after chemotherapy/before cystectomy and during post-cystectomy surveillance. They demonstrated that ctDNA-positive patients before and after NAC had significantly higher recurrence rates after cystectomy (HR = 29.1, *p* = 0.01 and HR = 12.0, *p* < 0.01, respectively). In addition, ctDNA status post-NAC was demonstrated to have a predictive role for pathological response before cystectomy, as ctDNA positivity was associated with residual tumour or positive lymph nodes at cystectomy specimens. On the other hand, all pT0 patients at cystectomy had previously undetectable ctDNA [10]. On a subsequent analysis of the same cohort with longer follow-up data (the median follow-up was 68 months), Lindskrog et al. confirmed the previously reported findings. ctDNA status, at all times before NAC (recurrence-free survival-RFS: HR = 15.6, 95% CI: 3.5–69; OS: HR = 8.9, 95% CI: 2.9–27.3), after NAC (RFS: HR = 15.2, 95% CI: 5–46.8; OS: HR = 9, 95% CI: 3.6–22.6) and after cystectomy (RFS: HR = 37.7, 95% CI: 8.5–167.1; OS: HR= 19.5, 95% CI: 6.9–54.6), remained highly prognostic of recurrence-free survival (RFS) and overall survival (OS). In addition, undetectable ctDNA before or after NAC was associated with an increased possibility of pathological downstaging before cystectomy and improved clinical outcomes [30]. Finally, Christensen et al. again utilised a tumour-informed approach with whole-exome sequencing of the primary tumour, to detect ctDNA before and after NAC in 91 patients treated with radical cystectomy. This cohort incorporated the aforementioned study [10]. They managed to validate their previous results suggesting a predictive role of ctDNA status for response to NAC by demonstrating that ctDNA positivity before or after NAC was associated with a lower pathological response rate to NAC and reduced RFS (*p* < 0.01) [31].

In addition, Szabados et al. carried out an exploratory analysis on the data of the ABACUS trial, based on the ctDNA status of the patients. This was a single-arm prospective study, which assessed the pathological response of MIBC following neoadjuvant treatment with the PD-L1 inhibitor atezolizumab. They assessed ctDNA on peripheral blood, following a tumour-informed approach with whole-exome sequencing of the tumour, in order to create patient-specific multiplex PCR assays. Of the patients, 63% had detectable ctDNA at baseline (before atezolizumab), 47% were ctDNA-positive post neoadjuvant treatment and 14% remained positive after cystectomy. A ctDNA-negative status at baseline or clearance post-atezolizumab were associated with a pathological response in the cystectomy specimen and with excellent prognosis with no relapses post-cystectomy. In addition, in patients with partial or complete pathological responses, ctDNA levels were reduced with atezolizumab, compared to non-significant changes in ctDNA levels in the non-responders. Finally, ctDNA-positive patients post-cystectomy were at a significantly higher risk of relapse, compared to patients with undetectable ctDNA levels (RFS HR = 78, *p* < 0.001) [32].

Similar findings were published by Van Dorp et al. as part of the NABUCCO trial (NCT03387761), in which good pathological response rates were reported in patients with locally advanced (stage III) MIBC treated with a combination of ipilimumab/nivolumab. A negative ctDNA status before cystectomy in two independent cohorts of the study was associated with response to combined immunotherapy (*p* < 0.01) and increased progression-free survival (PFS HR = 10.4, 95% CI: 2.9–37.5) [33].

Finally, in the first relevant study demonstrating the prognostic and predictive role of ctDNA published in 2017, Patel et al. reported a cohort of 17 patients treated with NAC, followed by cystectomy. They used a tumour-informed approach to detect ctDNA in peripheral blood and urine at different time points before NAC and after NAC, as well as before cystectomy and after cystectomy. They reported that baseline ctDNA levels before NAC were not prognostic or predictive of the response to NAC, but detectable ctDNA during NAC was associated with an increased risk of recurrence. The sensitivity and specificity of ctDNA-positive status at the second NAC cycle in predicting future recurrence was 83% and 100%, respectively, with a median time of recurrence at 293 days [34].

## 5. Circulating Tumour DNA (ctDNA) in the Post-Cystectomy and Adjuvant Setting

Seven studies evaluated the significance of post-cystectomy ctDNA levels as a marker of MRD, risk factor of recurrence and overall prognosis. Powles et al. published two updates of their exploratory analysis on data collected in the IMvigor010 (NCT02450331) study. This phase 3 randomised controlled trial (RCT) assessed the effect of the PD-L1 inhibitor atezolizumab versus observation on disease-free survival (DFS) in patients with high-risk MIBC post-radical cystectomy [35]. Although the study failed to demonstrate a statistically significant benefit of atezolizumab on DFS in the unselected population included, some interesting findings were demonstrated in the subsequent analysis, based on the ctDNA status of the patients. The authors used a tumour-informed approach with whole-exome sequencing of the tumour to identify 16 patient-specific mutations. They subsequently created a personalised multiplex PCR assay to locate these mutations in peripheral blood samples. In their initial report of 581 patients (biomarker-evaluable population—BEP) with a follow-up of 21.9 months, it was demonstrated that patients with detectable ctDNA immediately post-radical cystectomy (*n* = 214, 37%) had worse DFS and OS, compared to patients with undetectable ctDNA levels (observation arm DFS: HR = 6.3, 95% CI: 4.45–8.92 and OS: HR = 8.0, 95% CI: 4.92–12.99). In addition, atezolizumab had a positive effect on DFS and OS for the patients with detectable ctDNA levels post-cystectomy, compared to observation (DFS: HR = 0.58, 95% CI: 0.43–0.79 and OS: HR = 0.59, 95% CI: 0.41–0.86). This positive effect was not demonstrated in patients with undetectable ctDNA post-surgery (HR = 1.14, 95% CI: 0.81–1.62). Finally, ctDNA clearance at 6 weeks post-randomisation for the initially ctDNA-positive patients was observed in 18.2% of the patients in the atezolizumab arm, compared to 3.8% in the observation arm (*p* = 0.024) and was associated with improved DFS and OS, compared to the patients with persistently detectable ctDNA (DFS: HR = 0.26, 95% CI: 0.12−0.56) [36].

On a subsequent analysis of the same population with updated OS data and a median follow-up of 46.8 months, Powles et al. confirmed that ctDNA positivity immediately after cystectomy was associated with worse OS (median OS = 14.1 months, 95% CI: 10.5–19.7 vs. not reached, NE; HR 6.3, 95% CI: 4.3–9.3). In addition, patients with detectable ctDNA at baseline had a significantly better OS with adjuvant atezolizumab, compared to observation (median OS: 29.8 months, 95% CI: 20.7–40.2 vs. 14.1 months, 95% CI 10.5–19.7; HR 0.59, 95% CI: 0.42–0.83) but this positive effect was again not demonstrated in ctDNA-negative patients (HR =1.38, 95% CI: 0.93–2.05). Moreover, the authors demonstrated a quantitative correlation between the reduction in ctDNA levels with atezolizumab and OS, with patients experiencing complete ctDNA clearance having the best OS, followed by those with 50–99% and <50% reduction in ctDNA levels, respectively (60.0 months, 95% CI: 35.5–NE vs. 34.3 months, 95% CI: 15.2–NE vs. 19.9 months, 95% CI: 16.4–32.2). Finally, ctDNA positivity at baseline or 6 weeks post randomisation demonstrated a 57–68% sensitivity in identifying future radiological relapses 2.8–3.9 months in advance [21].

Smaller single-arm studies, also, demonstrated the potential role of ctDNA in monitoring MRD post-cystectomy and identifying high-risk patients for disease progression. Carrasco et al. published two studies investigating the prognostic value of cfDNA levels and ctDNA status post-radical cystectomy at different time points. In their first study of 37 patients, ctDNA status was assessed by using a tumour-informed approach with targeted next generation sequencing of the primary tumour, followed by digital droplet PCR of the peripheral blood samples. In a median follow-up of 36 months, 46% of the patients experienced disease progression. Increased cfDNA levels were identified as an independent predictor of progression (HR 5.290; *p* = 0.033) and positive ctDNA status as an independent predictor of reduced cancer-specific survival at 4 months after cystectomy (HR = 4.199, *p* = 0.038). In addition, they demonstrated that ctDNA positivity precedes radiological recurrence by a median of 6 months (*p* = 0.024), whereas ctDNA clearance 4 months post-cystectomy was associated with improved clinical outcomes (*p* = 0.033) [37]. On a subsequent study, the same group explored the feasibility and predictive value of a tumour-agnostic approach, aiming to introduce a simplified protocol that would omit prior tumour analysis. They utilised digital droplet PCR to identify two pre-defined mutations and reported a 24% progression rate in their cohort of 42 patients in a median follow-up of 21 months. They demonstrated that ctDNA-positive status before cystectomy and 4 and 12 months afterwards was associated with a higher risk of disease progression (HR 6.774, HR 3.673 and HR 30.865, respectively; *p* < 0.05). Finally, similarly to their previous study, they demonstrated that ctDNA clearance 4 months post-cystectomy was associated with longer progression-free survival, compared to the patients with persistently detectable ctDNA (*p* = 0.045) [38].

In addition, Birkenkamp-Demtroder et al. followed a tumour-informed approach with exome sequencing of the primary tumour to create digital droplet PCR patient-specific assays. They enrolled 60 patients, with half of them experiencing disease progression at a median of 275 days post-cystectomy. The median time between ctDNA detection and radiological progression was 101 days. ctDNA levels after cystectomy were significantly higher in the patients experiencing recurrence, compared to the group of recurrence-free patients (*p* < 0.001) [9]. In addition, in their cohort of 68 patients treated with NAC and cystectomy, Christensen et al., reported that a ctDNA-positive status post-cystectomy was highly prognostic of recurrence, with 76% of ctDNA-positive patients experiencing a recurrence by the end of the study follow-up (median follow-up 21 months), compared to no recurrences in the ctDNA-negative patients (*p* < 0.01). The median lead time between positive ctDNA status and radiological recurrence was 96 days [10]. Finally, on their updated analysis of the same cohort with a longer follow-up of 68 months, Lindskrog et al. reported that, by the end of the follow-up, 94% of the ctDNA-positive patients post-cystectomy had a recurrence, compared to only 4.2% of the ctDNA-negative patients. The sensitivity and specificity of the ctDNA-positive status were 89% and 98%, respectively. Finally, the prognostic value of ctDNA at diagnosis, before cystectomy (RFS HR = 3.4, 95% CI: 1.7–6.8) and after RC (accumulated ctDNA status RFS HR = 17.8, 95% CI: 3.9–81.2) was reproduced in a separate cohort of 102 NAC-naïve patients who were analysed in retrospect. RFS survival rates did not correlate with the history of NAC in the ctDNA-negative patients [30].

## 6. Conclusions—Future Directions

MIBC prognosis is currently poor and has essentially remained static over the last few decades. The modest survival benefit reported with the cisplatin-based NAC is compromised by the associated toxicities and only benefits a relatively small proportion of patients [7]. Several tissue and liquid biomarkers have been investigated in order to assess response to treatment, stratify patients according to the risk of progression/recurrence and guide the early start of systemic therapy before radiological recurrence [39,40].

The presence of certain mutations in tumour specimens has been associated with muscle invasive disease, an increased risk of progression/recurrence and a worse overall prognosis. TP53 gene mutations and protein p53 levels have been extensively assessed in urothelial malignancies and have been associated with an advanced clinical stage and a worse prognosis, although there is still some inconsistency in the reported results [41,42]. Similarly, levels of other histological biomarkers, such as p16 and Ki-67, have been associated with disease stage and the risk of progression. Hasan et al. reported that Ki-67 expression was associated with higher grade disease and muscle invasion, whereas the negative immunostaining of p16 was correlated with lower grade disease in papillary tumours [43]. In addition, there is a growing body of studies assessing the epigenetic changes on nucleic acids of bladder cancer patients. Recently, DNA methylation patterns have been shown to have diagnostic and predictive value in bladder cancer patients, with Lu et al. demonstrating that certain “DNA methylation signatures” in cell-free DNA predict the response to cisplatin-based NAC [44]. Finally, attention has been recently given to exogenic risk factors, such as infection by human papillomavirus (HPV), involved in the pathogenesis, clinical behaviour and progression of bladder cancer. A recent meta-analysis by Sun et al. demonstrated an increased risk of disease progression in the presence of HPV infection, in comparison to patients who were clear of HPV [45]. However, further studies are needed to shed light on the prognostic and predictive role of HPV status in relation to disease stage, other confounding risk factors and response to therapeutic options.

Most of the studies assessing prognostic and predicting factors in bladder cancer, and particularly in MIBC, have utilized tissue analysis from the initial transurethral resection of bladder tumour (TURBT) or cystectomy specimens. Therefore, they only investigate certain time points during the evolution of the disease and are mostly restricted to the primary tumours as source of tissue material [41,42,43]. Liquid biopsies represent a novel approach that allows for easy, non-invasive sampling and sequential, real-time assessment at different time points and in relation to different medical or surgical treatments. They do offer an overall assessment of the tumour burden, not restricted to the primary lesion, and can detect new mutational evolutions that can have significant therapeutic and prognostic implications [22,24].

There is a rapidly growing body of evidence supporting the role of ctDNA in different stages of the treatment of non-metastatic MIBC since the initial publications in 2017. Six studies reported a correlation between detectable ctDNA levels before cystectomy and increased risk of relapse or worse DFS/OS [10,29,30,32,33,34]. This finding might indicate metastatic disease at the time of surgery below the detection threshold of imaging techniques. However, ctDNA-negative patients were, also, reported to have recurrences, although significantly less often than ctDNA-positive patients. This might be due to a detection limitation of plasma ctDNA with the current lab methods and is likely to improve as more sensitive methods will be introduced in future clinical practice.

In addition, ctDNA levels can provide predictive information regarding the pathological response to cisplatin-based NAC or neoadjuvant immunotherapy. Three studies reported on ctDNA status in the context of cisplatin-based NAC and one study in the context of neoadjuvant atezolizumab. A ctDNA-negative status before cystectomy was shown to be associated with pathological downstaging after NAC to non-invasive stage or pT0, whereas a persistent ctDNA-positive status was shown to be related with a residual tumour of a higher stage in the cystectomy specimens [10,30,31]. Although the level of evidence is still low, this could mean that it would be possible to delay or even spare radical surgery in the future for patients who will be deemed to have a pathological complete response based on liquid biopsy monitoring and conventional surveillance methods. This would save a proportion of MIBC patients from the high perioperative complication rates and the life-changing consequences of urinary diversion.

Finally, seven studies evaluated ctDNA levels as a marker for MRD that could guide further escalation of treatment after radical cystectomy. These studies reported a strong correlation between positive ctDNA status post-cystectomy and an increased risk of recurrence and worse DFS/OS. It was also demonstrated that ctDNA positivity precedes radiological recurrence by 2.8–6 months, and thus offers a window for earlier adjuvant treatment with potential survival benefits. In fact, as demonstrated by the explanatory analysis of the data from the IMvigor010 trial, the ctDNA-positive patients can experience survival benefits with adjuvant immunotherapy, in contrast to the unselected post-cystectomy population [21,36].

This is one of the very few review articles that summarized evidence published to date assessing the prognostic and predictive role of ctDNA in non-metastatic MIBC treated with cystectomy with or without neoadjuvant/adjuvant treatments. Overall, the outcomes reported in the included studies are very encouraging. Further advances in laboratory techniques, which will allow for the more sensitive and less costly analysis of ctDNA, are expected to further highlight the utility of liquid biopsies. In particular, ctDNA appears to be a very promising novel biomarker that could change the current oncological management of MIBC from treating unselected cohorts of patients to an individualized approach, based on the risk of progression or degree of response to treatments. This will open new horizons in the new era of precision oncology, with potential implications for other urological and non-urological malignancies. However, the current level of evidence published in the literature is still low, as it mainly consists of non-comparative studies with small populations or secondary analyses of RCTs with other primary endpoints. Therefore, larger, dedicated RCTs are needed to confirm and validate the utility of liquid biopsies in the management of non-metastatic MIBC.

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
