# Peer review of "Circulating Tumour DNA and Its Prognostic Role in Management of Muscle Invasive Bladder Cancer: A Narrative Review of the Literature"

_biomedicines, 2024, doi:10.3390/biomedicines12040921_

Round 1

Reviewer 1 Report

Comments and Suggestions for Authors

Manuscript review- biomedicines-2960958

Thank you for the opportunity to review this interesting study: manuscript biomedicines-2960958 entitled: “Circulating tumour DNA and their prognostic role in management of muscle invasive bladder cancer".

The authors have conducted a narrative review of the current literature aiming to present the utilization of ctDNA in different crossroads in the management of non-metastatic MIBC.

The authors focus on the neoadjuvant, perioperative and post-cystectomy settings.

The manuscript is well written and structured and is supported by the most updated literature. Hence, I recommend it to be published in your esteem journal.

Author Response

Dear Editor and Reviewers,

We would like to thank you again for the opportunity to contribute to your journal on this rather interesting and novel topic that is likely to revolutionise the management of urothelial cancers. We would also like to thank the three reviewers for the very kind and encouraging comments regarding our first draft and for the time they dedicated to critically review it.

We are very grateful for the recommendations made by the reviewers and for the two interesting references they attached for our consideration. We have added and discussed both of them in the “Conclusion- Future Directions” section (lines 274-288). We have also added a paragraph to further highlight the novelty of the liquid biopsies and particularly ctDNA in the management of non- metastatic MIBC as requested (lines 289-300). We have marked both paragraphs red for easier identification. `Finally, we have added the phrase “narrative review” in the title of the manuscript as recommended and also marked it red and reviewed the whole manuscript for linguistic errors, which were corrected.

We would be more than happy to further revise the manuscript if required and we look forward to hearing back from you.

Yours sincerely,

Lazaros Tzelves

MD, MSc, PhD, FEBU, ECFMG

Assistant Professor of Urology

National and Kapodistrian University of Athens, Greece

Reviewer 2 Report

Comments and Suggestions for Authors

Congratulations, it is a very well-conducted narrative review concerning an area of interest that is becoming more important day by day. 

Review:

Title: Circulating Tumour DNA and Its Prognostic Role in Management of Muscle Invasive Bladder Cancer

This narrative review of the literature on liquid biopsy in muscle-invasive bladder cancer aims to provide an overview of the current studies on ctDNA, highlighting its prognostic and predictive role in the management of MIBC. This work contributes significantly to the literature by gathering and synthesizing the strengths and weaknesses of the studies on this topic to date, focusing on the predictive power of this biomarker.

No areas of weakness in the work were detected by the reviewer. The narrative review appears complete, clear, and comprehensive regarding the subject matter. The topic is more relevant than ever, and an attempt is being made to bridge the existing gap in the literature with this manuscript, which serves to provide an overview of the situation concerning ctDNA and its use in the management of MIBC.

The conclusions and future directions summarize the most relevant findings of the studies considered, emphasizing how such a promising biomarker could truly change the management of patients with MIBC, towards a personalized treatment of selected patient cohorts.

Author Response

(The authors gave the same response as above.)

Reviewer 3 Report

Comments and Suggestions for Authors

Major comments:

The novelty of the topic, authors should clarify

Minor comments:

-Authors should provide (a narrative review) in the title

- More details about recent associations between invasive carcinoma and other factors such as HPV should be given: I recommend using these recent sources:  https://doi.org/10.3390/clinpract13040073 & https://doi.org/10.1002%2Fjmv.28208

Author Response

(The authors gave the same response as above.)
